# Experimental study on the effect of Coumarone resin on the performance of SBS-modified asphalt

**Chenyu Feng**[1], **Chunhua Hu**[1,2]*, **Zhaobin Sun**[1], **Hao Zhang**[1], **Zhaozhao Xu**[1]

**1** School of Civil Engineering, Architecture and Environment, Hubei University of Technology, Wuhan, China,
**2** Key Laboratory of Intelligent Health Perception and Ecological Restoration of Rivers and Lakes, Ministry of Education, Hubei University of Technology, Wuhan, China

* hu_chunhua@163.com

**Data Availability Statement:** All relevant data are within the manuscript and its Supporting Information files.

**Funding:** The author(s) received no specific funding for this work.

## Abstract

An inexpensive and high-performing solid Coumarone resin was added to Styrene-butadiene-styrene (SBS) copolymer-modified asphalt to enhance its storage stability and road performance. To assess the effect of Coumarone resin dosage on the SBS-modified asphalt, a series of laboratory tests were conducted. The composite modified asphalt's segregation test was used to evaluate its storage stability, Dynamic Shear Rheometer (DSR) and Multiple Stress Creep Recovery (MSCR) tests were employed to investigate its high-temperature performance and permanent deformation resistance, and the Bending Beam Rheology (BBR) test was utilized to measure its low-temperature performance. Fluorescence microscopy was used to observe the composite modified asphalt's microstructure, and Fourier Transform Infrared Spectroscopy (FTIR) was conducted to study the changes in chemical structure during the modification process. The results showed that Coumarone resin can improve the compatibility of SBS and asphalt, improve the high-temperature performance and deformation resistance of SBS-modified asphalt, and adding an appropriate amount of Coumarone resin can help enhance the low-temperature cracking resistance of modified asphalt. The optimal dosage of Coumarone resin recommended for SBS-modified asphalt performance enhancement is 2% under the test conditions, as determined by comparing the test results of samples with various dosages.

## 1. Introduction

With the rapid development of China's economy, asphalt pavements play an increasingly important role in the road network [1]. Although asphalt pavements have the advantages of excellent load-bearing capacity, fast construction speed and high reparability, they are also prone to cracks, depressions and rutting in service, leading to a reduction in life expectancy [2]. Asphalt is crucial for road pavement, therefore improving its performance is a major issue. It comprises four fractions: saturated hydrocarbons, aromatic hydrocarbons, resins, and asphaltenes, collectively known as SARA fractions [3]. Modifiers are typically used to alter the SARA component and structure of asphalt to improve its viscoelastic properties and overall performance.

**Competing interests:** The authors have declared that no competing interests exist.

Styrene-butadiene-styrene (SBS) copolymers are commonly used to enhance the performance of asphalt due to their superior high-temperature, low-temperature, and aging resistance properties [4–6]. When SBS and asphalt are mixed together, due to the relatively large differences in their respective structures and compositions, there are bound to be compatibility problems. The compatibility of SBS with asphalt refers to the ability of the two to accommodate each other and form a macroscopically homogeneous modified asphalt [7]. Moreover, modified asphalt is prone to phase separation, leading to poor storage stability during transportation and storage. Additionally, an increase in SBS content during mixing and construction can result in elevated asphalt viscosity, making it difficult for pumping and mixing, which is not beneficial for construction [8]. This is because SBS is more compatible with aromatic hydrocarbons and colloids in asphalt, but less compatible with saturated hydrocarbons and asphaltenes. Therefore, it is essential to incorporate aromatic-rich materials as a compatibilizer to enhance the compatibility of SBS with the matrix asphalt. Currently, there are more studies on oil substances as compatibilizers, such as furfural extraction oil, aromatic oil, shale oil, etc. However, the addition of oil substances will reduce the consistency of matrix asphalt, which ultimately affects the rutting resistance of SBS-modified asphalt [9–11]. Petroleum resin with nine carbon atoms ($C_9$ petroleum resin), as a malleable hydrocarbon resin made by polymerization of petroleum cracking products, has some double bonds and aromatic rings in its structure, which makes it compatible with other resins, rubbers, SBS and matrix bitumen [12]. Existing research indicates that $C_9$ petroleum resin enhances the compatibility between SBS and asphalt without compromising the high-temperature performance of the asphalt. It serves as an optimal compatibility modifier for SBS-modified asphalt [13–16]. Nevertheless, petroleum resources are finite, and the continuous escalation of petroleum prices erodes the cost advantage of related products.

Coumarone resin to Ethylene tar, carbon nine, and other heavy aromatic hydrocarbon components by the high-temperature polymerization reaction of the resin, the composition is mainly a short side chain necking thick ring aromatic hydrocarbons. It exhibits excellent compatibility with rubber, SBS, and SIS while offering advantages such as cheapness, sustainability, heat resistance, and aging resistance [17]. Yang SQ et al. demonstrated that the use of Coumarone resin as an activator significantly improved the compatibility of rubber powder and asphalt [18]. Huang WD et al. developed a high-viscosity asphalt modifier based on Coumarone resin and oleic acid amide as raw materials [19]. Pyshyev S et al. added gumarone resin to base bitumen, the softening point of bitumen was increased from 47°C to 52°C, and the bonding property was also significantly improved, but meanwhile, the Coumarone resin also reduced the plastic properties of bitumen, especially the needle penetration [20, 21]. In a subsequent study, Pyshyev S suggested the use of tar with Coumarone resin as a modifier for bitumen [22]. Shved M et al. investigated the effect of Coumarone resins containing methacrylic fragments on the main properties of bitumen and showed that Coumarone resin doped at 1% had little effect on the ductility of bitumen but doubled the adhesion of bitumen [23]. Lin X et al. separately employed solid Coumarone resin and liquid Coumarone resin as modifiers for asphalt, with the findings indicating that solid Coumarone resin notably enhances the high-temperature performance of asphalt, while liquid Coumarone resin enhances its low-temperature performance [24]. Current scholars have mostly studied the effect of Coumarone resin on base asphalt, but the effect of Coumarone resin on SBS-modified asphalt has not been reported. For this reason, this experiment will be Coumarone resin added to SBS-modified asphalt, to explore the effect of Coumarone resin on the physical properties and high and low temperature performance of SBS-modified asphalt. This is achieved through the preparation of Coumarone resin/SBS composite-modified asphalt. In comparison to SBS-modified asphalt, a battery of tests encompassing fundamental performance evaluations, temperature-frequency

**Table 1. Main performance indexes of matrix asphalt.**

| Test items | Test results | standardized value |
|---|---|---|
| Dissolution rate (%) | 99.7 | — |
| Penetration (25˚C, 0.1mm) | 62 | 60–80 |
| Softening point (˚C) | 47.8 | $\geq$45 |
| Ductility (15˚C /cm) | 102 | $\geq$100 |
| Flash point (˚C) | 310 | — |

scanning tests, multiple stress creep recovery tests (MSCR), and low-temperature beam bending rheological tests (BBR) are employed to investigate the physical properties, rheological behavior, and storage stability of the composite-modified asphalt. Additionally, fluorescence microscopy tests and Fourier-transform infrared spectroscopy (FTIR) are employed to examine the microstructure and modification mechanisms of the composite-modified asphalt. This study is geared towards enhancing the storage stability, high-temperature stability and low-temperature crack resistance of SBS-modified asphalt while expanding the application of Coumarone resin in the realm of asphalt.

# 2. Materials and tests

## 2.1 Materials

The asphalt used in this study is a 70# matrix asphalt produced by domestic refineries, as detailed in Table 1 for its performance indicators. SBS adopts linear SBS791 from Shenzhen Jitian Chemical Co., Ltd., its performance indicators are shown in Table 2; Coumarone resin adopted C-140 of Shanghai Xingxu Industrial Co.,Ltd., its performance indicators are shown in Table 3. Based on the results of preliminary pre-tests and existing references [25–27], SBS dosage based on engineering commonly used dose, that is 4% of the dosage of asphalt (mass fraction, the same below), indicating that the ordinary roads modified asphalt [28], and the dosage of the Coumarone resin was 0%, 1%, 2%, 3%, and 4% of the asphalt mass, which were denoted as $CR_0$, $CR_1$, $CR_2$, $CR_3$, and $CR_4$, respectively (the same below).

## 2.2 Sample preparation

Method for preparing Coumarone resin/SBS-modified asphalt: Initially, the asphalt underwent heating to a temperature range of 175˚C to 185˚C. Following this, the asphalt was introduced to the SBS modifier, constituting 4% of the asphalt mass. The ensuing process involved slow stirring, subsequent addition of varying quantities of Coumarone resin, and high-speed shearing at a shear rate of 5,000 r/min. Stirring was consistently maintained throughout the entire preparation procedure, with the temperature carefully controlled within the range of 175˚C to 185˚C. The overall shear duration encompassed 40 minutes, measured from the moment Coumarone was introduced until the completion of shearing [29]. Following shearing, the asphalt

**Table 2. Main performance indices of linear SBS791.**

| Test items | Test results |
|---|---|
| Block ratio (S/B) | 30/70 |
| Tensile strength (MPa) | 18 |
| Stretch elongation (%) | 815 |
| Shore hardness (A) | 75 |
| Melt Flow Index (g/10min) | 4 |

**Table 3. Technical indices of Coumarone resin.**

| Test items | Test results |
| --- | --- |
| Softening point (°C) | 146 |
| Color number | 12~18 |
| Acid value (%) | ≤0.5 |
| Ash (%) | ≤0.1 |
| Bromine value (mg) | ≤30 |

temperature was maintained within the range of 150°C to 160°C for about 2 hours. Stirring was consistently applied during this duration to aid in the full expansion of the SBS modifier.

## 2.3 Testing program

To investigate and characterize the basic properties, rheological properties and microstructure of the modified asphalt prepared above, the tests including penetration, softening point, ductility, temperature frequency scanning, bending beam rheometer (BBR), fluorescence microscopy and fourier transform infrared spectroscopy (FTIR) were performed in this study. The flowchart of testing program is shown in Fig 1.

**2.3.1 Basic performance test.** To investigate the impact of Coumarone resin on the efficacy of SBS-modified asphalt, we conducted assessments on the physical characteristics of Coumarone resin-altered SBS-modified asphalt. These evaluations adhered to the specifications outlined in the "American Society of Testing Materials" (ASTM), specifically D5-06, D36-95, D113-99, and D5329. The parameters examined encompassed penetration, softening point, ductility at 5°C, and elastic recovery at 25°C. Among them for the warm areas of SBS modified asphalt penetration should be in the range of 35~55 (0.1mm), the softening point should be greater than 60°C, ductility at 5°C should be greater than 200 mm, the elastic recovery rate of not less than 70%.

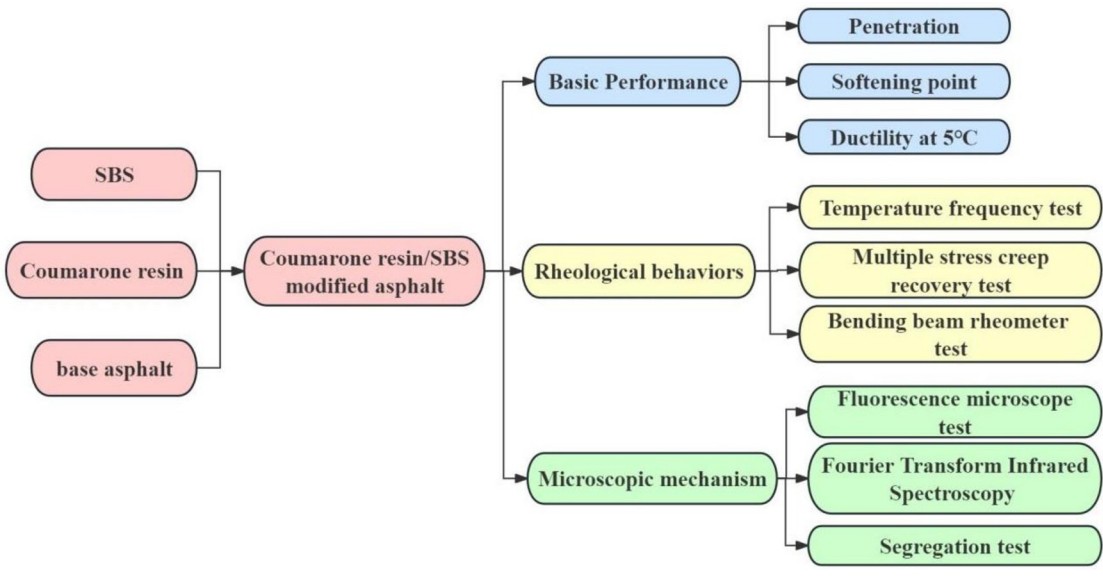

**Fig 1. Flowchart of testing program.**

**2.3.2 Temperature frequency test.** In order to investigate the rheological properties of the modified asphalt prepared above under high and medium temperatures, a dynamic shear rheometer (DSR, Malvern Kinexus Pro+, Malvern Panalytical Ltd, UK) was used in this study and testing procedures provided in AASHTO T315-12 [34] were followed.

The frequency scanning tests on asphalt samples, aiming to quantify the complex shear modulus (G*) and phase angle (δ) of the modified asphalt, were executed using a DSR. The chosen configuration involved parallel plates with a 25 mm diameter and a 1 mm spacing between the axial plate and the bottom plate. The examination was carried out across a temperature spectrum of 40~100°C with a temperature gradient of 5°C. Strain loading at 0.5% was applied, and the scanning frequency ranged from 0.1 to 100 rad/s.

**2.3.3 Multiple stress creep recovery (MSCR) test.** The MSCR test is a widely employed method for assessing the resistance to permanent deformation in polymer-modified asphalt. To characterize the high-temperature performance of the modified asphalt, parameters such as Elastic Recovery (R) and Unrecoverable Creep Flexibility ($J_{nr}$) are employed. The test adhered to the American Society for Testing and Materials (ASTM D7405) protocol. Firstly, the asphalt was aged, and a rotating film oven (RTFOT) was often used to simulate short-term ageing in the experiments [30]. 35g of unaged asphalt was packed into glass bottles, which were placed into the turntable inside the RTFOT equipment, and the turntable was rotated at a speed of 15 ±0.2r/min for 85min at 163±0.5°C to obtain the short-term aged asphalt. Modified asphalt specimens, subjected to short-term aging, underwent ten intermittent creep stress cycles at 0.1 kPa and 3.2 kPa sequentially at 64°C, with a cycling period comprising 1s of creep and 9s of recovery. The Jnr-diff, indicative of stress sensitivity in asphalt, was derived by calculating $J_{nr}(0.1)$ and $J_{nr}(3.2)$ using the formula outlined in Eq 1.

$$J_{nr-diff} = [(J_{nr}(3.2) - J_{nr}(0.1))/J_{nr}(0.1)] \times 100\% \tag{1}$$

Where: $J_{nr}$ (0.1) and $J_{nr}$ (3.2) are the unrecoverable creep flexibility at standard stress levels of 0.1 kPa and 3.2 kPa, respectively. Jnr-diff is an indicator of stress sensitivity.

**2.3.4 Bending beam rheometer (BBR) test.** According to AASHTO T313, BBR tests were conducted on SBS-modified asphalt with different Coumarone dosages to evaluate the low-temperature performance of the binder [31]. The creep rate (m) and creep stiffness (S) of the asphalt binders at different temperatures (-12°C and -18°C) were obtained by placing the beams (127 mm×12.7 mm×6.35 mm) in a water bath at a certain temperature for 60 min, and then applying a constant load of 980±50 mN for 240s in the beams. Two replicate tests were performed for each sample and the average value was taken as the test result.

**2.3.5 Fluorescence microscope test.** The mechanism of SBS-modified asphalt reinforced with Coumarone resin was investigated by fluorescence microscopy test, a small amount of specimen was placed on a slide, covered with a coverslip, and gently pressed on a hot stage at 163°C, then observed and photographed with a Nikon AIR-type laser confocal microscope [32]. Through the different transmittance between the modifier and asphalt, the distribution state of the SBS modifier in asphalt can be observed, thus evaluating the compatibility between the SBS modifier and asphalt.

**2.3.6 Fourier Transform Infrared Spectroscopy (FTIR) analysis.** In this study, the functional groups of the modified bitumen were analyzed using attenuated total reflectance spectroscopy (FTIR-ATR) as a way to investigate whether there is a chemical reaction during the modification of SBS bitumen by Coumarone resin. The specimens were first dissolved in carbon tetrachloride ($CCl_4$) for no less than 2 h. The mass ratio of asphalt to $CCl_4$ was 1:10, and after the asphalt was completely dissolved, the solution was dripped onto a potassium bromide (Kbr) slide with a micro-syringe, and then the $CCl_4$ solution was evaporated with an infrared

lamp. The scanning frequency was set at 32 times/min for this test, and the wavelength of the infrared spectrum was 500~4000 cm$^{-1}$.

**2.3.7 Segregation test.** Assess the storage stability of modified asphalt through a segregation test following the specifications outlined in ASTM D7173-14: (1)Dispense 50g of the asphalt specimen into an aluminum tube with a diameter of 25.4mm and a height of 140mm. Position the tube vertically in an oven at 163˚C for 48 hours. (2)Promptly remove the aluminum tube from the oven and transfer it to a refrigerator set at -5˚C for a cooling period of 4 hours. (3)Remove the aluminum tube, and put it onto a flat plate. The specimen was evenly divided into three sections for assessing the softening point at both the top and bottom. The smaller the variation in softening points between the upper and lower ends, the higher the storage stability of the modified bitumen.

## 3. Results and discussion

### 3.1 physical property

The physical property data of Coumarone resin/SBS composite modified asphalt at different dosages of Coumarone resin are shown in Fig 2. The figure illustrates that the addition of Coumarone resin led to a 21% decrease in the penetration of the modified asphalt. The reason is that in SBS, the PS segment determines the strength of the material, the resin molecules into the PS segment in SBS can increase the force between PS segment and asphalt [15], thereby enhancing the consistency of SBS modified asphalt.

The softening point of the asphalt modified with Coumarone resin/SBS composite showed an escalation corresponding to the increase in Coumarone resin dosage. Specifically, SBS-CR$_4$ experienced an 11% elevation in its softening point. This is attributed to the inherent high softening point of C-140 Coumarone resin and its property to enhance polymer cohesion, reinforcing the bond between SBS and asphalt. Consequently, this creates a mesh structure within the asphalt, ultimately improving the high-temperature performance of the modified asphalt.

The 5˚C ductility of the Coumarone resin/SBS composite modified asphalt demonstrates a pattern of initially increasing and subsequently decreasing with the elevation of Coumarone resin dosage. Notably, the ductility of SBS-CR$_2$ experiences a remarkable improvement of 48%. This occurrence arises from the polymer with unsaturated double bonds present in the Coumarone resin. Consequently, SBS assimilates an ample amount of lightweight components, thereby augmenting the low-temperature flexibility of the modified asphalt. It's important to note that excessive use of Coumarone resin, being a brittle material with poor low-temperature performance, can diminish the modified asphalt's resistance to low-temperature cracking, leading to reduced ductility.

### 3.2 High-temperature performance

Temperature-frequency scanning of various asphalts using a dynamic shear rheometer (DSR) yields complex shear modulus (G*) and phase angle (δ), which can be used to reflect the viscoelastic properties of asphalt materials.

G* represents the ratio of maximum shear stress to maximum shear strain and indicates a material's resistance to deformation under specific shear loads [33]. A higher G* value in asphalt signifies better deformation resistance, particularly at high temperatures. Fig 3 illustrates the temperature-dependent variation in the G* value for Coumarone resin/SBS composite-modified asphalt. The observed trend indicates a decline in G* for each asphalt sample as the temperature rises. This reduction in G* is attributed to heightened internal asphalt temperatures, which lead to the attenuation of intermolecular forces. Consequently, the asphalt undergoes a transition from a highly elastic state to a more viscous flow state, resulting in

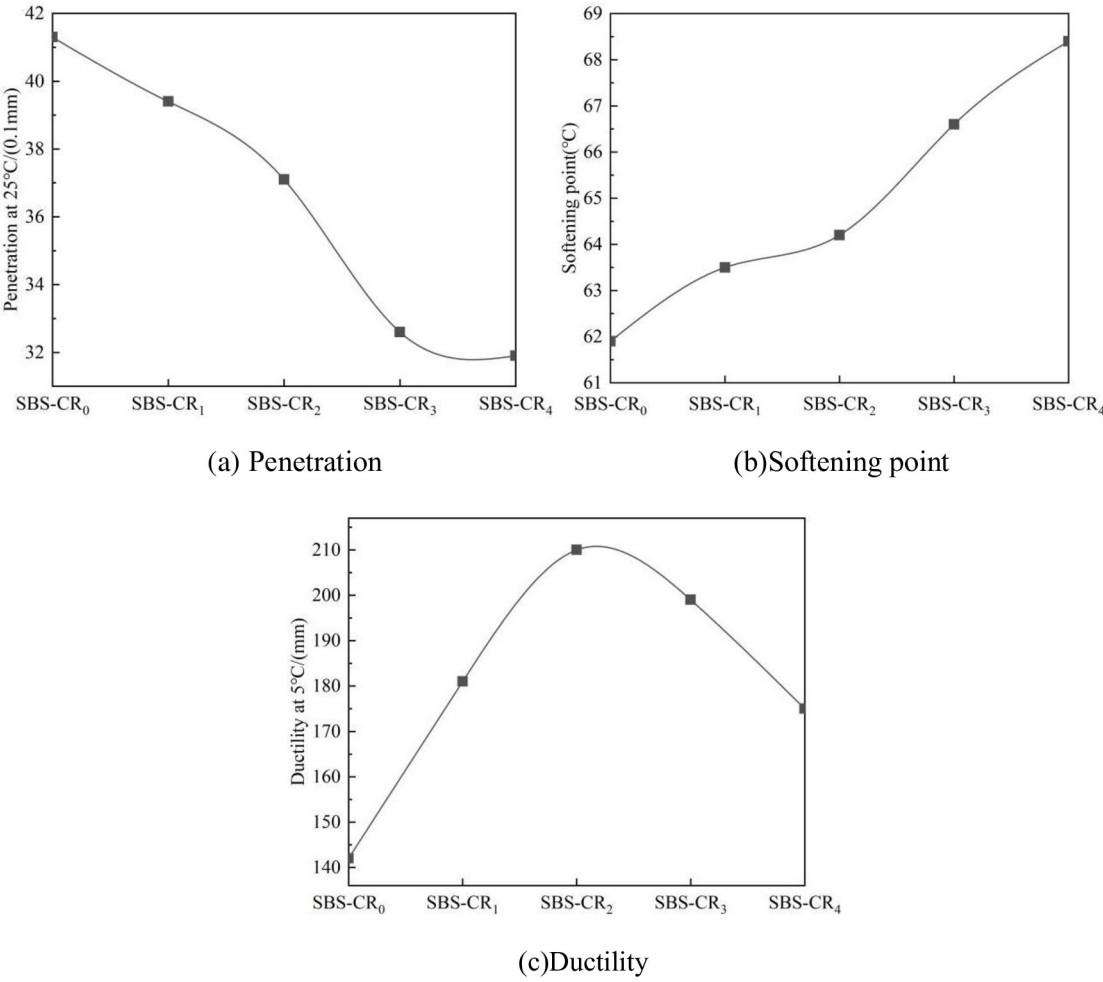

(a) Penetration                                    (b)Softening point

(c)Ductility

**Fig 2. Influence of Coumarone resin dosage on the physical properties of SBS asphalt.** (a) Penetration, (b) Softening point, (c) Ductility.

diminished resistance to deformation [34]. Furthermore, at the same temperature, the $G^*$ of the modified asphalt increases progressively with higher dosages of Coumarone resin. For instance, at 64°C, the $G^*$ of SBS-CR$_4$ is approximately 2.3 times that of SBS-CR$_0$. The substantial improvement in stress resistance of SBS-modified asphalt is attributed to the inclusion of Coumarone resin. This is achieved through the role of Coumarone resin in reducing the size of SBS particles during the preparation of Coumarone resin/SBS composite modified asphalt. Smaller particles are more effectively dissolved in the asphalt, resulting in a denser mesh structure between SBS and asphalt. Consequently, this improves the resistance of the modified asphalt to deformation.

δ reflects the change of the viscous and elastic components in the asphalt material, which holds significance in comprehending asphalt behavior. Asphalt as an elastic-plastic material, a smaller δ value indicates a higher proportion of elasticity, signifying greater deformability recovery. Conversely, a larger δ value indicates the presence of a significant amount of irreversible deformation within the asphalt. δ is more sensitive to changes in the internal physical and chemical structure of asphalt than the $G^*$ value and more accurately reflects changes in the viscoelasticity of asphalt materials with temperature [35]. In Fig 4, the temperature-dependent

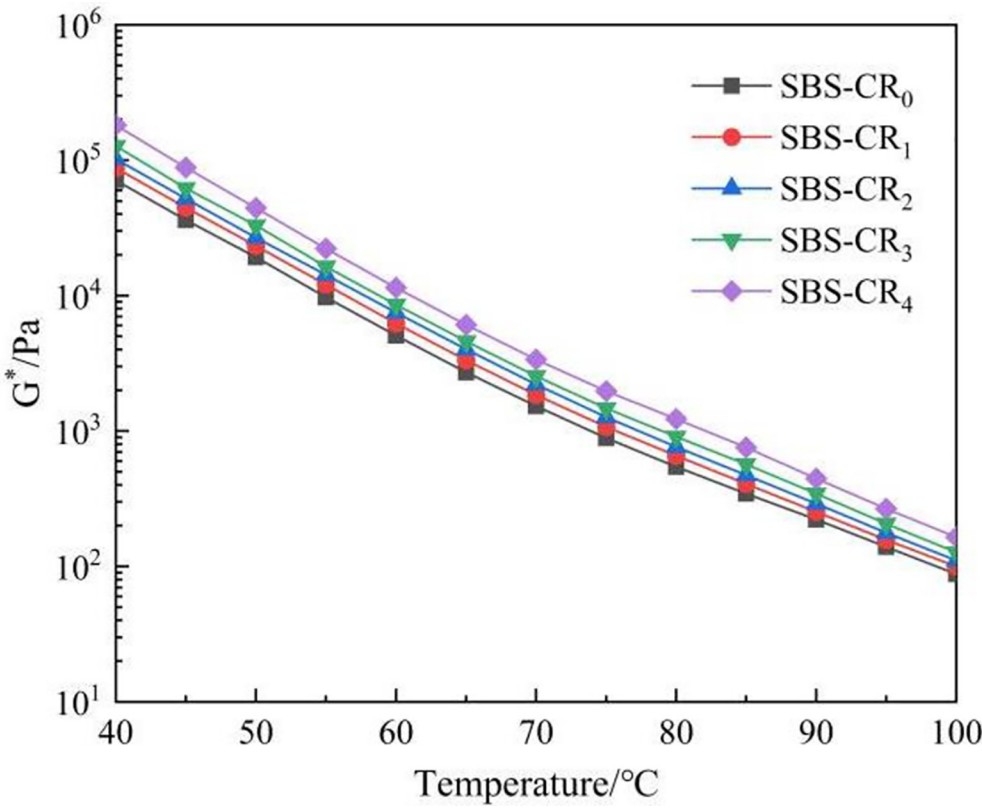

**Fig 3. Complex shear modulus($G^*$).**

variation curve of the δ-value for Coumarone resin/SBS composite-modified asphalt is depicted. Notably, the incorporation of Coumarone resin results in a conspicuous reduction in the phase angle of the modified asphalt. This implies a decreased temperature sensitivity of the modified asphalt, particularly in the elevated temperature range spanning from 50˚C to 85˚C. It is also observed that the phase angle of the modified bitumen reaches its minimum at 2% of Coumarone resin dosage and thereafter increases with the increase in Coumarone resin dosage. This phenomenon is credited to the capability of Coumarone resin, when employed at an optimal dosage, to modify the proportion of viscoelastic components within the modified asphalt. That is, the proportion of saturated fraction and gelatine in the modified asphalt is reduced, resulting in the largest relative molecular mass of asphaltene as the centre, around which adsorbed the most polar gelatine, the asphalt formed at this time for the gel-type asphalt, with excellent elasticity. Nonetheless, it should be noted that Coumarone resin inherently exhibits polymer-like viscosity characteristics, and an excessive application in the modified asphalt can have adverse effects on viscosity and elasticity.

Rutting resistance factor ($G^*/\sin\delta$) is used to evaluate the ability of asphalt materials to resist deformation under repetitive loading. The U.S. SHRP study introduced the $G^*/\sin\delta$ value to indicate the high-temperature performance of asphalt mixtures [36]. A larger $G^*/\sin\delta$ value indicates greater stability at high temperatures, signifying stronger rutting resistance. As illustrated in Fig 5, the rutting resistance factor $G^*/\sin\delta$ of Coumarone resin/SBS composite modified asphalt exhibits a trend parallel to that of $G^*$. Specifically, $G^*/\sin\delta$ in modified asphalt increases as the dosage of Coumarone resin rises within the temperature range of 40~100˚C. For instance, SBS-CR₄ displays a significant increase of 2950 Pa at 64˚C when compared to

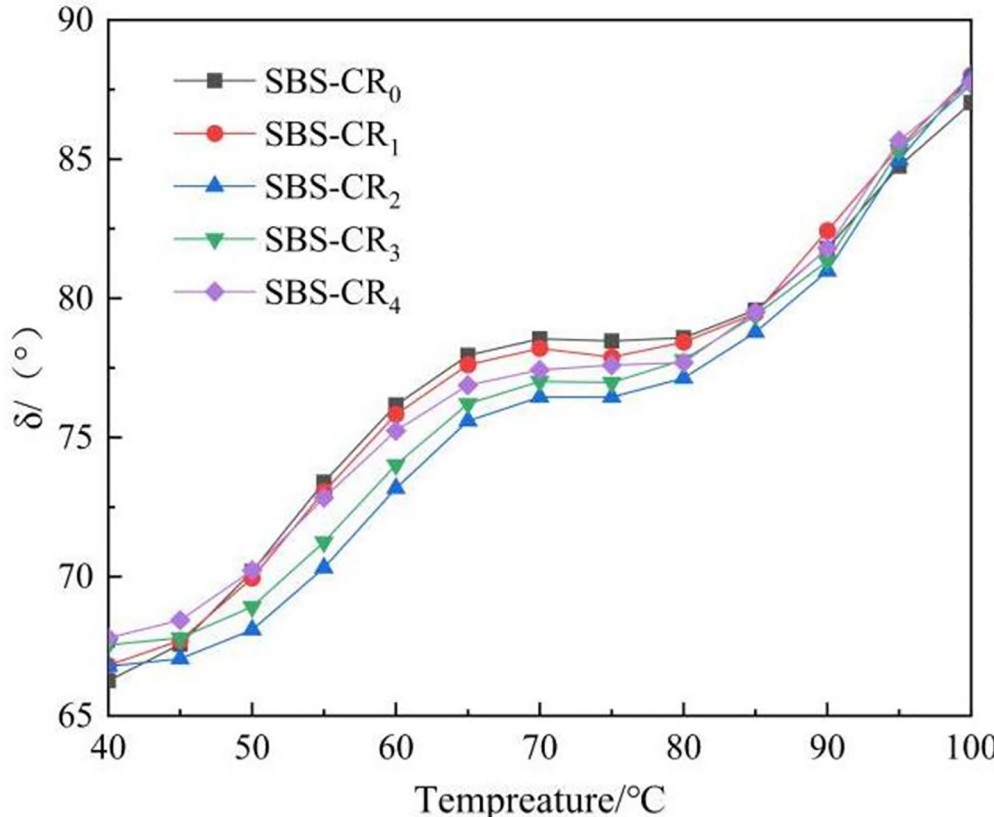

**Fig 4. Phase angle (δ).**

SBS-CR$_0$. This can be attributed to the fact that Coumarone resin reduces the particle size of the SBS particles, thereby increasing the contact area with the asphalt. This, in turn, enhances the cohesion of the modified asphalt, leading to improved rutting resistance [37].

## 3.3 Low-temperature performance

The susceptibility of asphalt to low-temperature cracking can be assessed through the Bending Beam Rheology (BBR) test. Test outcomes provide values for creep strength (S) and creep coefficient (m). The S value serves as an indicator of the asphalt's flexibility at low temperatures, while the m value signifies the stress relaxation capacity of the asphalt. Enhanced low-temperature cracking resistance is indicated by a smaller S-value and a larger m-value in modified asphalt. Adhering to the AASHTO T313-04 specification, optimal performance is achieved when S is less than 300 MPa, and m-values exceed 0.3. This study delves into the low-temperature performance of various modified asphalts at -12˚C and -18˚C, with the findings presented in Fig 6.

It's apparent that, as the temperature decreases, the same modified asphalt exhibits an increase in the S value and a decrease in the m value. This trend indicates a deterioration in the cracking resistance of the modified asphalt at lower temperatures [38]. And with the increase of Coumarone resin blending, the S-value of SBS-CR$_2$ decreased by 33% at -12˚C and 54% at -18˚C; the m-value of SBS-CR$_2$ increased by 0.028 at -12˚C and 0.027 at -18˚C. This suggests that the inclusion of Coumarone resin improved the low-temperature cracking resistance of SBS-modified bitumen. The rationale behind this enhancement lies in the abundance

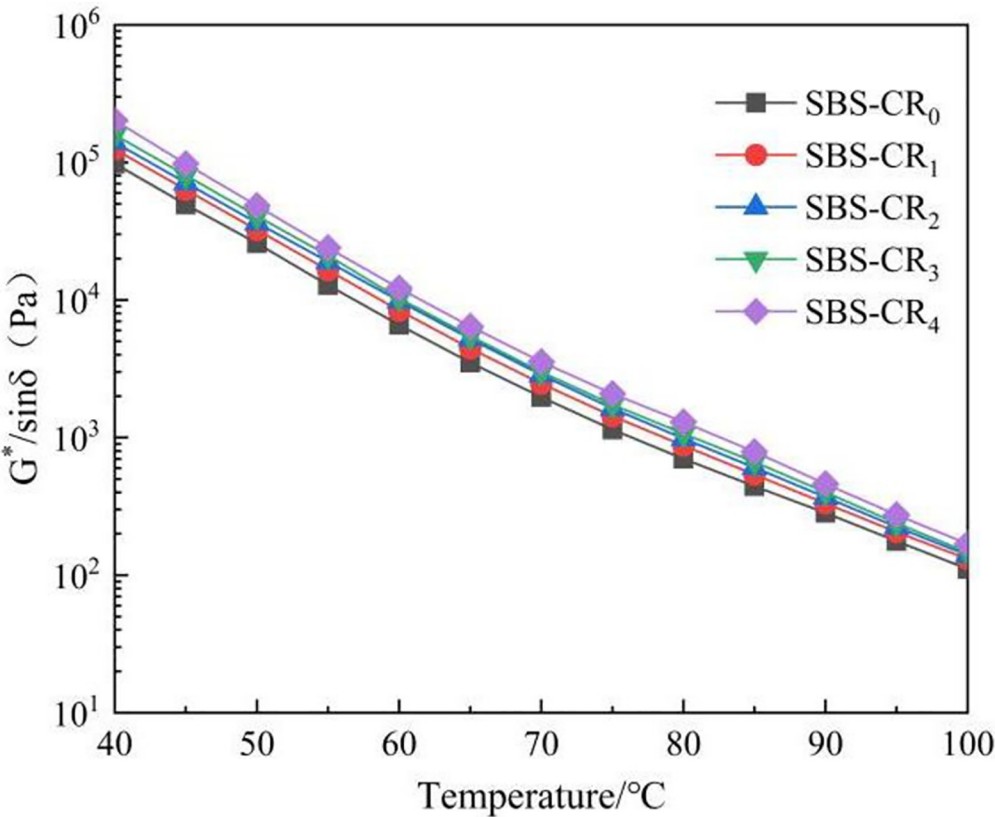

**Fig 5. Rutting resistance factor (G\*/sinδ).**

of lightweight components in Coumarone resin, which enhances the compatibility between SBS and asphalt. As a result, more effective crosslinking occurs between SBS and asphalt phases, forming a robust three-dimensional mesh structure that enhances the resistance to low-temperature cracking performance. This reduces or avoids stress concentration due to SBS particle aggregation under external loading, which can lead to cracks in the asphalt and low-temperature damage. However, when the Coumarone resin dosage reaches 3%, the Coumarone resin adversely affects the low-temperature performance of its composite modified asphalt with SBS. This correlation aligns with the findings from the earlier 5°C ductility test. It implies that while Coumarone resin has a positive impact on the low-temperature performance of asphalt by promoting the formation of a new structure, an overuse of Coumarone resin may contribute to heightened asphalt brittleness due to the inherent brittleness of the resin itself. This, in turn, diminishes asphalt's stress relaxation properties, resulting in a decline in low-temperature cracking resistance.

### 3.4 Resilience against deformation

The SHRP program proposed the Multiple Stress Creep Recovery (MSCR) test to further evaluate the deformation recovery of asphalt. The MSCR test examines the viscoelastic deformation of asphalt under various stress levels. Asphalt deforms under stress loading, where elastic deformation is recoverable creep deformation and viscous deformation is unrecoverable creep deformation and accumulates to the next creep loading stage, the test can more realistically simulate the process of asphalt pavement subjected to loads in service [39]. The MSCR test

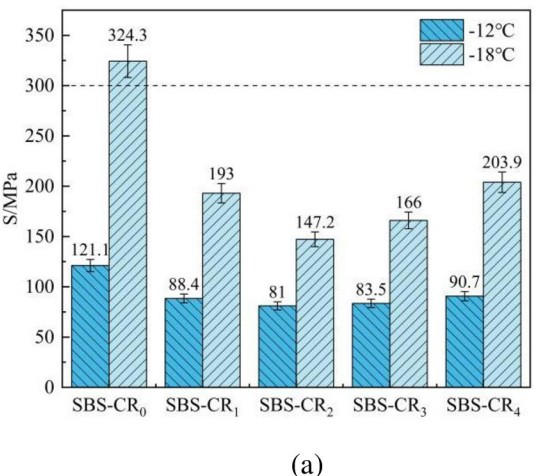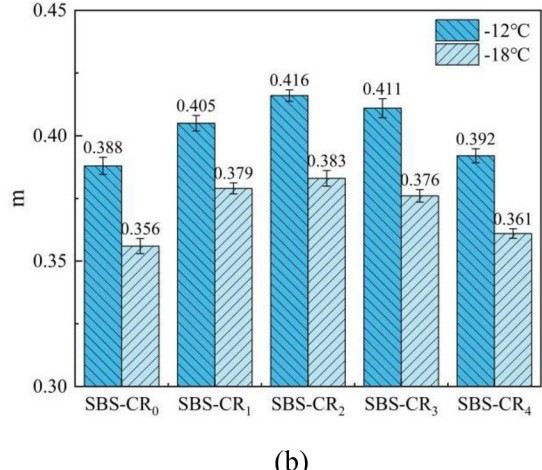

(a)                                                                (b)

**Fig 6. Variation of S and m with temperature.**

employs a stress-control model to examine the unrecoverable flexibility ($J_{nr}$) of the modified asphalt, focusing on stress levels of 0.1 kPa and 3.2 kPa. A smaller Jnr value signifies a heightened capability of the asphalt to rebound from deformation at the specified temperature. The test commenced by applying a stress of 0.1 kPa, followed by the execution of 10 recovery cycles. Subsequently, the stress was increased to 3.2 kPa, and an additional 10 cycles were conducted, concluding the test after a total duration of 200 seconds. This sequence of steps was executed at the outset of the testing procedure.

The results of the MSCR test are presented in Fig 7. As the temperature gradually increases, the Jnr of the modified asphalt exhibits a consistent upward trend. For example, when subjected to stresses of 0.1 kPa and 3.2 kPa, the Jnr of SBS-CR$_0$ experienced increments of 198% and 199%, respectively. This underscores the parallel impact of temperature and stress, indicating that alterations in either factor can result in deformation damage in modified asphalt. The maximal damage to the modified asphalt occurs when both high temperature and high stress are concurrently present. As the concentration of Coumarone resin doping increased,

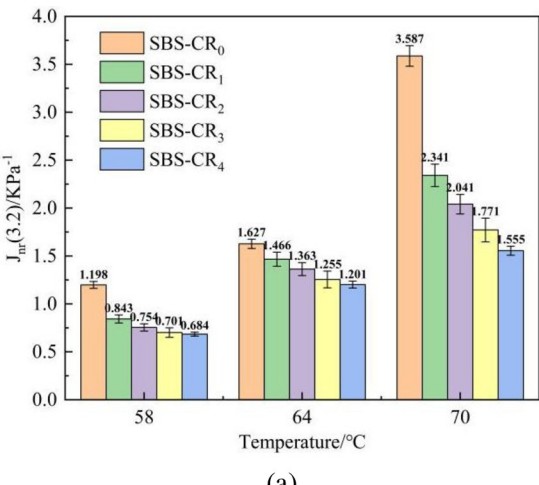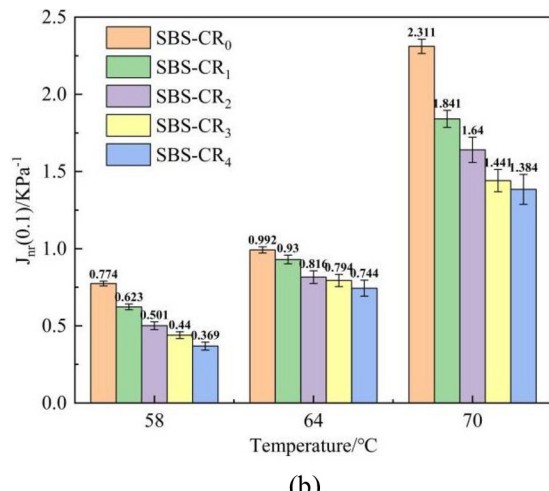

(a)                                                                (b)

**Fig 7. $J_{nr}$-value of Coumarone resin/SBS composite modified asphalt.**

there was a notable decrease in both $J_{nr}(0.1)$ and $J_{nr}(3.2)$ at various temperatures, where the $J_{nr}(0.1)$ and $J_{nr}(3.2)$ of SBS-CR$_4$ at 64˚C decreased by 25% and 26%, respectively, comparing with that of SBS-CR$_0$. This is attributed to the augmentation of the elastic component in the modified asphalt facilitated by the presence of Coumarone resin. As a result, the modified asphalt exhibits improved resistance to permanent deformation at elevated temperatures. This is due to the fact that the lightweight component provided by the Coumarone resin meets the absorption of the SBS polymer and can reach saturation. The SBS particles in this state are in contact with each other to form an elastic whole, which enables the modified asphalt to recover better from the deformation caused by the load after deformation.

The $J_{nr\text{-}diff}$-value serves as a metric to mirror the extent of stress sensitivity in asphalt. This parameter is a crucial indicator for assessing the distinct stages of rheological properties in asphalt [40]. $J_{nr\text{-}diff}$-values in the range of ≤5.0%, 5.0% to 75.0%, >75.0%, corresponding to the linear region, non-linear region (has not yet occurred creep damage), and creep damage stage. Solving for $J_{nr\text{-}diff}$ provides a comprehensive understanding of asphalt deformation across different temperature ranges. The stress sensitivity of the Coumarone resin/SBS composite-modified asphalt is shown in Fig 8. Notably, only the Jnr-diff of SBS-CR$_4$ at 58˚C surpasses 75%, while its Jnr-diff at 70˚C gradually decreases with increased Coumarone resin dosage. This suggests that Coumarone resin effectively alleviates the stress sensitivity of the modified asphalt, particularly at elevated temperatures. Regarding the AASHTO MP-19 specification, the $J_{nr}(3.2)$ and $J_{nr\text{-}diff}$ are used as indicators to classify into four classes, namely Extremely Heavy Traffic (E), Very Heavy Traffic (V), Heavy Traffic (H), and Standard Traffic (S), as shown in Table 4.

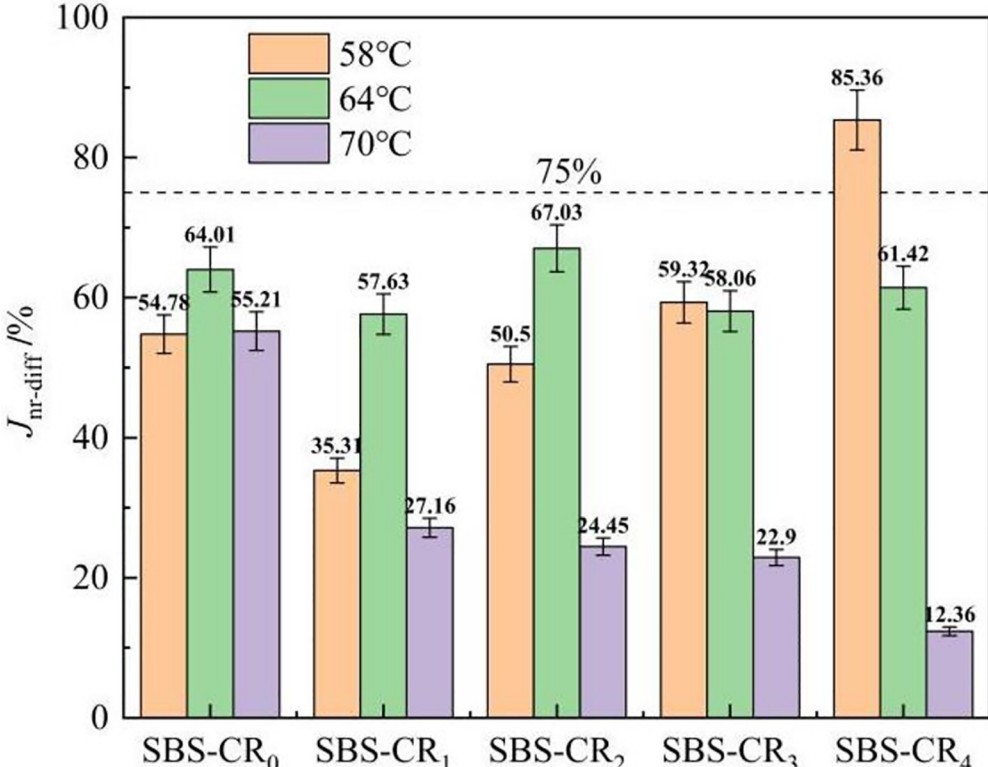

**Fig 8. $J_{nr\text{-}diff}$ test results of Coumarone resin/SBS composite modified asphalt.**

**Table 4. MSCR test grading criteria.**

| Traffic classification | $J_{nr}$ (3.2)/kPa$^{-1}$ | $J_{nr\text{-}diff}$ /% |
|---|---|---|
| S(Standard Traffic) | ≤4.0 | ≤75% |
| H(Heavy Traffic) | ≤2.0 | ≤75% |
| V(Very Heavy Traffic) | ≤1.0 | ≤75% |
| E(Extremely Heavy Traffic) | ≤0.5 | ≤75% |

At 58°C, modified asphalts with more than 1% Coumarone resin were able to achieve extra heavy traffic (V), and at high temperatures up to 70°C, modified asphalts with 3% and 4% Coumarone resin were still able to achieve heavy traffic (H). This implies that incorporating Coumarone resin into SBS asphalt enhances the applicability of the modified asphalt to various traffic classifications. The main reason is due to the better compatibility of Coumarone resin with petroleum asphalt and thermoplastic rubber. This compatibility helps to promote cross-linking between the SBS polymer and asphalt. Coumarone resin also has excellent aging resistance, weathering resistance, acid and alkali resistance, and a high softening point. These characteristics greatly enhance the modified asphalt's elastic component at high temperatures, improving its resistance to permanent deformation and ability to recover from deformation.

## 3.5 Microstructure

The interplay between the SBS polymer and asphalt significantly influences the performance of SBS-modified asphalt. Investigating the microstructure of Coumarone resin/SBS composite-modified asphalt is instrumental in elucidating the mechanism through which Coumarone resin enhances the performance of SBS-modified asphalt. Fig 9 shows fluorescence micrographs (magnified 200 times) at different Coumarone resin dosages. In the images, SBS polymers appear as bright spots with yellow fluorescence, while Coumarone resin and asphalt present a dark background. From Fig 9(A), it can be observed that there is a significant difference in particle sizes between SBS particles, ranging from 10 to 100μm and unevenly dispersed. The reason for this is that there are differences in relative molecular mass, density, polarity, and solubility parameters between SBS polymers and bitumen, and both of them just have weak van der Waals forces, and it is difficult to form a stable mixed system by conventional physical dispersion. Therefore the different physical properties and molecular numbers cause incomplete mutual solubility between the two [7]. From Fig 9(B), it can be found that the introduction of 1% Coumarone resin results in a notable reduction in SBS particle size, with particles below 50μm and a more uniform distribution. In Fig 9(C), the SBS particle size continues to decrease, resembling the distribution in Fig 9(B), but with more SBS particles appearing at the same scale. It is shown that the increase in Coumarone resin blending provides more lightweight components and significantly improves the compatibility and dispersion of SBS polymers in asphalt. Finally, in Fig 9(D) and Fig 9(E) the SBS particles are in the form of uniformly sized scattering points, all of which have a particle size of less than 20μm, and form a reticulated structure with the SBS particles as the backbone. This is because the Coumarone resin increases the asphaltene content in the asphalt, condensing more and larger agglomerates of asphaltene wrapped by gelatine in the bitumen, thus increasing the force between the bitumen and SBS to prevent agglomeration between SBS particles, so that SBS can be sheared to a smaller particle size.

The fluorescence microscope test demonstrates that in the preparation of composite modified asphalt, Coumarone resin effectively enhances the shearing efficiency of SBS particles, resulting in the absorption and dissolution of smaller SBS particles in the asphalt, forming a

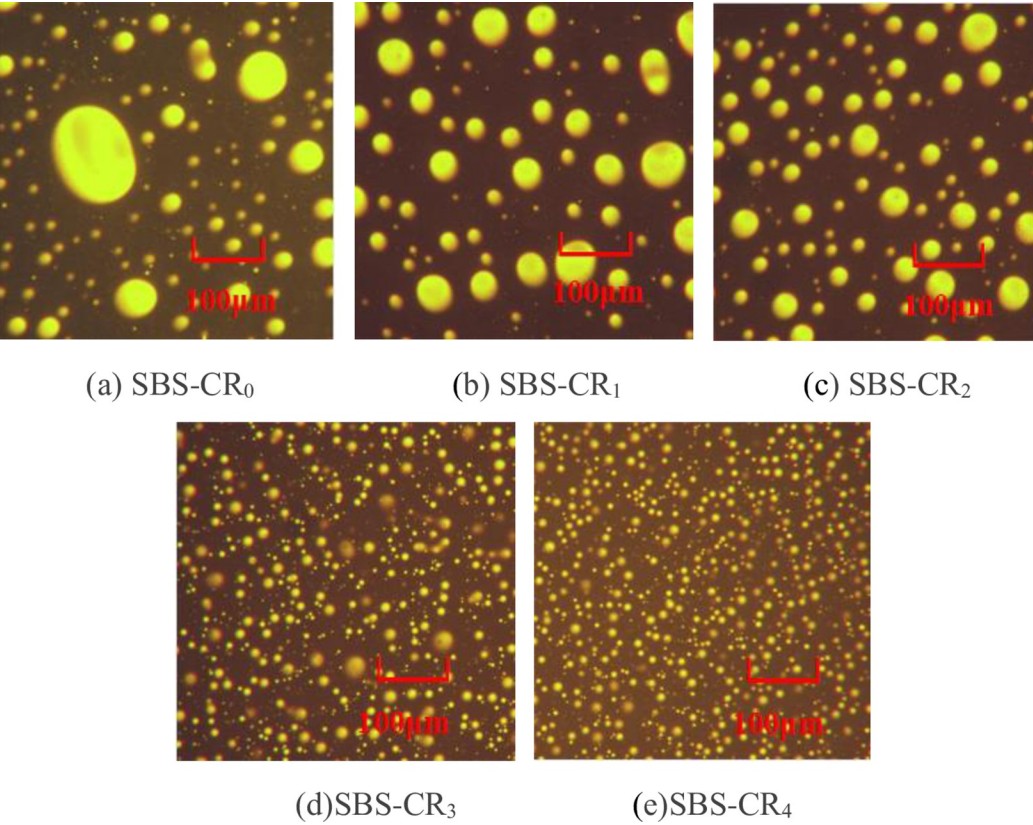

**Fig 9. Fluorescence microscope images of Coumarone resin/SBS composite modified asphalt.** (a) SBS-CR$_0$, (b) SBS-CR$_1$, (c) SBS-CR$_2$, (d)SBS-CR$_3$, (e)SBS-CR$_4$.

closely linked structure. This explains how Coumarone resin enhances the high-temperature performance and deformation resistance of SBS-modified asphalt [41].

## 3.6 Modification mechanism

Infrared spectroscopy is capable of capturing transmittance (or absorbance) change curves across various wavelengths for material molecules. Observing differences in the positions of absorption peaks can provide information about chemical bonds or relevant functional groups within material molecules. This analytical technique helps ascertain whether a chemical reaction occurs during the modification process [42]. In this study, FTIR tests were conducted on virgin asphalt, SBS-CR$_0$, and SBS-CR$_2$ to explore and analyze their molecular compositions.

Different asphalt infrared spectroscopy results are shown in Fig 10. It can be seen that the infrared spectra of SBS-CR$_0$, SBS-CR$_2$ and base asphalt are very similar. Some of the absorption peaks are only slightly different in intensity, there are no new characteristic absorption peaks or the original characteristic peaks disappeared. It was observed that the absorption peaks at 2923cm$^{-1}$ and 2855cm$^{-1}$ increased after the addition of the Coumarone resin, which was caused by the -CH$_2$ antisymmetric and symmetric telescoping vibration. This suggests that the addition of the Coumarone resin enhanced the bonding energy of the methylene group and thus the molecular force between the asphalt and the SBS. There is a less obvious absorption peak at 1599cm$^{-1}$, which is generated by the stretching vibration of the C = C aromatic ring, due to the high bond energy of the C = C double bond, and its rise indicates that

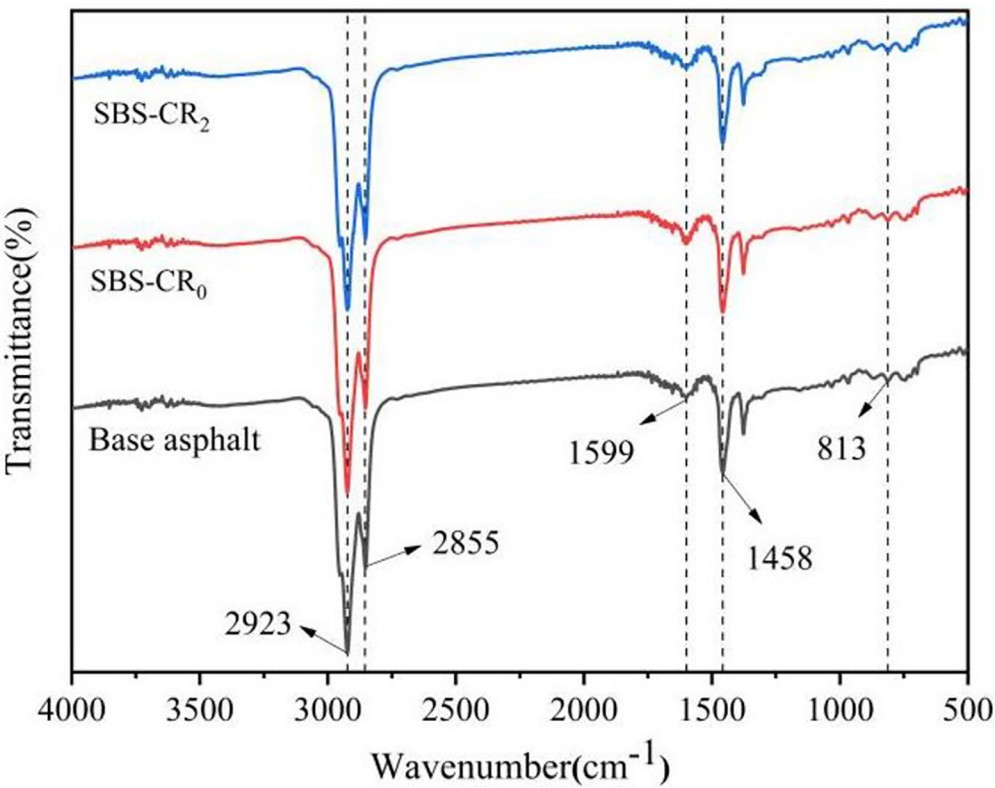

**Fig 10. Infrared spectra of Coumarone resin/SBS composite modified asphalt and base asphalt.**

the mechanical properties of the Coumarone resin/SBS composite modified asphalt are enhanced, This corresponds to the high resistance to deformation of the Coumarone resin/SBS composite modified asphalt in the rheological performance tests; The absorption peak at $1458cm^{-1}$, on the other hand, is caused by the shear vibrational absorption of $-CH_3$; The wavelength range of 910–650 $cm^{-1}$ corresponds to the benzene ring substitution region of asphalt. Within this range, several weak absorption peaks arise from the vibration of the C-H bond on the benzene ring. The infrared spectroscopy test indicates that there are no additional absorption peaks in the Coumarone resin/SBS-modified asphalt composite compared to the base asphalt. This observation indicates that the process of preparing modified asphalt, accomplished by blending Coumarone resin and SBS, represents a physical modification instead of introducing novel chemical reactions.

### 3.7 Storage stability

The storage stability of modified asphalt holds substantial importance in its practical application. The thermodynamic incompatibility between the SBS polymer and asphalt, coupled with the restricted presence of lightweight components in the matrix asphalt, impedes the effective solubilization of SBS. Therefore, during storage at high temperatures, SBS particles will float and aggregate, while heavier components, such as asphaltenes, will sink, ultimately resulting in the phenomenon of segregation [43]. The segregation test was used to evaluate the storage stability of the modified asphalt and the results are shown in Table 5.

Clearly, as the dosage of Coumarone resin increased from 0% to 4%, there was a significant and observable rise in the softening point for both the upper and lower sections of the

**Table 5. Results of segregation test of Coumarone resin/SBS composite modified asphalt.**

| Test Asphalt | location | Softening point/°C | $\Delta T_{R\&B}$/°C |
| --- | --- | --- | --- |
| SBS-CR$_0$ | Top | 65.9 | 10.5 |
| | bottom | 55.4 | |
| SBS-CR$_1$ | Top | 66.5 | 3.9 |
| | bottom | 62.6 | |
| SBS-CR$_2$ | Top | 65.7 | 2.3 |
| | bottom | 63.4 | |
| SBS-CR$_3$ | Top | 67.8 | 1.5 |
| | bottom | 66.3 | |
| SBS-CR$_4$ | Top | 69.4 | 1.2 |
| | bottom | 68.2 | |

aluminum tubes. Moreover, the softening point values for the upper sections of the aluminum tubes exhibited an increase compared to the values measured prior to the segregation test. This is due to the fact that there is an upwelling of the SBS polymer in the asphalt, causing SBS particles to aggregate at the top of the aluminum tube. In addition, the difference between the softening point of the top and the bottom ($\Delta T_{R\&B}$) was significantly reduced with the addition of the Coumarone resin. This results from the incorporation of Coumarone resin, an aromatic hydrocarbon polymer, into the asphalt. It provides a lightweight component so that the chain segments of the SBS polymer can be stretched, reducing the free energy of its own surface, which can prevent the SBS polymer from agglomerating again, thus promoting the compatibility of the SBS polymer with the asphalt. The formation of SBS and asphalt two interpenetrating network structure, which restricts the mobility of the asphalt. The segregation test proved that the co-blended structure of the Coumarone resin/SBS composite modified asphalt is stable and not vulnerable to segregation under hot storage, which is in line with the results of the fluorescence microscopy test.

# 4. Conclusions

Through the preparation of Coumarone resin/SBS composite modified asphalt, using SBS-modified asphalt as a control group, this study aims to explore the effect of Coumarone resin on the performance of SBS-modified asphalt and the role of the mechanism, the test results are analyzed to obtain the following conclusions:

1. An appropriate quantity of Coumarone resin proves effective in enhancing the fundamental properties of SBS asphalt. In the baseline measurements, the incorporation of Coumarone resin led to a reduction in the penetration of SBS-modified asphalt. Specifically, SBS-CR$_4$ demonstrated a 6.5°C increase in softening point, a 20% improvement in elastic recovery, and a noteworthy 68mm expansion in 5°C ductility.

2. Findings from temperature frequency scanning tests indicate that the addition of Coumarone resin enhances the high-temperature deformation resistance of SBS-modified asphalt. The findings of the BBR test show diverse levels of enhancement in the low-temperature cracking resistance of SBS-modified asphalt, depending on the varying quantities of Coumarone resin. Nevertheless, it's noteworthy that when the Coumarone resin content surpasses 2%, the rheological properties of SBS-modified asphalt experience adverse effects.

3. The results of the separation test reveal that Coumarone resin effectively mitigates the floating and aggregation of SBS within the asphalt, thereby enhancing the storage stability of

asphalt modified with SBS. This improvement is attributed to the ability of Coumarone resin to supply a considerable number of lightweight components, subsequently increasing the compatibility between SBS and asphalt.

4. Through the fluorescence microscope can be found, that the Coumarone resin greatly reduces the particle size of SBS, makes its distribution more uniform, and forms a tight network structure, which is the important reason that the Coumarone resin on SBS asphalt properties have a variety of effects. FTIR analysis showed that the modification mechanism of Coumarone resin with SBS was through physical blending.

5. In summary, the recommended amount of Coumarone resin is 2% when combining all the properties as well as microanalysis. Finally, research on the pavement properties of Coumarone resin/SBS-modified asphalt composite aggregates needs to be further carried out. The dosage of SBS in this experiment was 4% to comply with the index of asphalt used in ordinary roads, and thus the dosage and preparation conditions of the obtained Coumarone resin are only suitable for pavements under ordinary conditions. In the advanced pavement and complex conditions, the mixing amount of SBS and reference index should be adjusted appropriately. In addition, since Coumarone resin is mostly used as a viscosity builder in the rubber field, there should be a great potential for future research in the preparation of high-viscosity asphalt.

## Supporting information

**S1 Data.**
(ZIP)

## Author Contributions

**Conceptualization:** Hao Zhang.

**Data curation:** Chenyu Feng.

**Formal analysis:** Chenyu Feng.

**Funding acquisition:** Chunhua Hu.

**Investigation:** Chenyu Feng.

**Methodology:** Hao Zhang.

**Project administration:** Zhaobin Sun.

**Resources:** Chenyu Feng, Zhaozhao Xu.

**Software:** Zhaobin Sun, Zhaozhao Xu.

**Validation:** Chunhua Hu.

**Writing – original draft:** Chenyu Feng, Chunhua Hu.

**Writing – review & editing:** Chenyu Feng, Chunhua Hu.

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
