## [Decision Letter · Decision Letter 0]

23 Jan 2024

PONE-D-23-42395Experimental study on the effect of Coumarone resin on the performance of SBS-modified asphaltPLOS ONE

Dear Dr. Hu,

Thank you for submitting your manuscript to PLOS ONE. After careful consideration, we feel that it has merit but does not fully meet PLOS ONE’s publication criteria as it currently stands. Therefore, we invite you to submit a revised version of the manuscript that addresses the points raised during the review process.

We look forward to receiving your revised manuscript.

Kind regards,

Mayank Sukhija

Academic Editor

PLOS ONE

Journal Requirements:

2. We note that your Data Availability Statement is currently as follows: "All relevant data are within the manuscript and its Supporting Information files."

4. Please upload a copy of Supporting Information (S1 DATE) which you refer to in your text on page 24 (in PDF format).

**Additional Editor Comments:**

Overall, the manuscript require major revision.

Appropriate reasoning is missing throughout the manuscript. Kindly provide the reasoning to support the results with suitable references.

Reviewers' comments:

Reviewer's Responses to Questions

**Comments to the Author**

1. Is the manuscript technically sound, and do the data support the conclusions?

Reviewer #1: Yes

Reviewer #2: Yes

2. Has the statistical analysis been performed appropriately and rigorously? 

Reviewer #1: Yes

Reviewer #2: No

3. Have the authors made all data underlying the findings in their manuscript fully available?

Reviewer #1: Yes

Reviewer #2: Yes

4. Is the manuscript presented in an intelligible fashion and written in standard English?

Reviewer #1: Yes

Reviewer #2: Yes

5. Review Comments to the Author

Reviewer #1: The authors have presented an experimental study to evaluate the effect of Coumarone resin on the performance of SBS-modified asphalt. The paper topic is relevant and interesting. The following comments are made to improve the paper:

1. Throughout the manuscript, the author shall use the same font for mentioning Degree Celsius.

2. L55: The authors used the abbreviation "C9", it is suggested to provide the full form for first time.

3. In Table 1, 2, and 3, it is recommended to provide all units in brackets '()', instead of using '/'

4. It is recommended to cite all the ASTM, AASHTO, and JTE specifications in proper reference style.

5. Section 2.8: Please provide chemical terms in proper form, i.e. In CCl4, 4 should be subscripted.

Reviewer #2: 1. Please check the manuscript again.

2. Add an experimental plan for the manuscript

3. Please add statistical comparison if possible, and error bars for all the bar charts in the manuscript.

4. The reasoning for the results is missing throughout the manuscript please add them.

6. PLOS authors have the option to publish the peer review history of their article (what does this mean?). If published, this will include your full peer review and any attached files.

Reviewer #1: No

Reviewer #2: **Yes: **Vipul Chitnis

---

## [Author Response · Author response to Decision Letter 0]

11 Feb 2024

Sincerely thank you for your letter and reviewers’ comments concerning our manuscript entitled “Experimental study on the effect of Coumarone resin on the performance of SBS-modified asphalt” (PONE-D-23-42395). These valuable comments are very helpful for us to revise and improve the quality of the paper. We have studied comments carefully and made corrections which we hope meet with approval. Since the article underwent major changes, we have revised the Introduction, test program and Discussion section. Also, Some of the tables, figures and acknowledgment have also been updated.

---

## [Decision Letter · Decision Letter 1]

5 Mar 2024

PONE-D-23-42395R1

Experimental study on the effect of Coumarone resin on the performance of SBS-modified asphalt

PLOS ONE

Dear Dr. Hu,

Thank you for submitting your manuscript to PLOS ONE. After careful consideration, we feel that it has merit but does not fully meet PLOS ONE’s publication criteria as it currently stands. Therefore, we invite you to submit a revised version of the manuscript that addresses the points raised during the review process.

**ACADEMIC EDITOR: Please do the needful as per the comments of the Reviewers. There are still chances of improvement in the paper. **

We look forward to receiving your revised manuscript.

Kind regards,

Mayank Sukhija

Academic Editor

PLOS ONE

Journal Requirements:

Reviewers' comments:

Reviewer's Responses to Questions

**Comments to the Author**

1. If the authors have adequately addressed your comments raised in a previous round of review and you feel that this manuscript is now acceptable for publication, you may indicate that here to bypass the “Comments to the Author” section, enter your conflict of interest statement in the “Confidential to Editor” section, and submit your "Accept" recommendation.

Reviewer #1: All comments have been addressed

Reviewer #2: All comments have been addressed

Reviewer #3: (No Response)

2. Is the manuscript technically sound, and do the data support the conclusions?

Reviewer #1: Yes

Reviewer #2: Yes

Reviewer #3: Yes

3. Has the statistical analysis been performed appropriately and rigorously? 

Reviewer #1: Yes

Reviewer #2: Yes

Reviewer #3: N/A

4. Have the authors made all data underlying the findings in their manuscript fully available?

Reviewer #1: (No Response)

Reviewer #2: Yes

Reviewer #3: Yes

5. Is the manuscript presented in an intelligible fashion and written in standard English?

Reviewer #1: Yes

Reviewer #2: Yes

Reviewer #3: Yes

6. Review Comments to the Author

Reviewer #1: (No Response)

Reviewer #2: Although all the comments are addressed carefully, kindly read the manuscript once again and fix the formatting discrepancies in the article. Eg: Please capitalize the word flowchart in figure 1. The heading numbers are not formatted according to the journal requirements, Section 3.1.

Reviewer #3: This study investigated the effect of Coumarone resin and SBS-modified asphalt’s performance using DSR, MSCR, BBR, and FTIR. In its current state, the paper has several shortcomings and needs reworks before it can be considered for publication. My primary concerns are as follows:

1. The introduction section should be revised: More literature on coumarone resin-modified asphalt is required.

2. LN 44 requires reference – “However, the compatibility between SBS and asphalt is limited”. In addition, what did the authors mean by “compatibility”. Please further elaborate.

3. LN 74-77: The research gap is unclear. Some studies already assess the coumarone resin performance when used in conjunction with asphalt. What does this study do differently?

4. LN85-87: what is road performance? Did you mean cracking resistance of flexible pavements? Or the rutting performance of AC? Please reframe the sentence for further clarity.

5. LN95: Why did the authors select 4% SBS ratio?

6. LN 126-127: Please provide the standards in references as well.

7. LN 145: Could you please provide details of short-term aging?

8. LN 156: Please provide references.

9. LN 191-194: How did authors conclude that the decrease in penetration is evidence of improvement in compatibility?

10. LN 221, please provide some references. Similarly, LN240-245. Also, to LN264.

11. Please also include limitations and future work in the manuscript.

7. PLOS authors have the option to publish the peer review history of their article (what does this mean?). If published, this will include your full peer review and any attached files.

Reviewer #1: No

Reviewer #2: **Yes: **Vipul Chitnis

Reviewer #3: No

---

## [Author Response · Author response to Decision Letter 1]

15 Mar 2024

Dear Editors and Reviewers:

Sincerely thank you for your letter and reviewers’ comments concerning our manuscript entitled “Experimental study on the effect of Coumarone resin on the performance of SBS-modified asphalt” (PONE-D-23-42395). These valuable comments are very helpful for us to revise and improve the quality of the paper. We have studied comments carefully and made corrections which we hope meet with approval.We have uploaded the specific modification details and look forward to your review again, Thank you!

---

## [Editor Report · Decision Letter 2]

18 Mar 2024

Experimental study on the effect of Coumarone resin on the performance of SBS-modified asphalt

PONE-D-23-42395R2

Dear Dr. Hu,

We’re pleased to inform you that your manuscript has been judged scientifically suitable for publication and will be formally accepted for publication once it meets all outstanding technical requirements.

An invoice for payment will follow shortly after the formal acceptance. To ensure an efficient process, please log into Editorial Manager at Editorial Manager® , click the 'Update My Information' link at the top of the page, and double check that your user information is up-to-date. If you have any billing related questions, please contact our Author Billing department directly at authorbilling@plos.org.

Kind regards,

Mayank Sukhija

Academic Editor

PLOS ONE

Additional Editor Comments (optional):

The paper is now in perfect shape and flow.
---

## [Editor Report · Acceptance letter]

8 Apr 2024

PONE-D-23-42395R2 

PLOS ONE

Dear Dr. Hu, 

I'm pleased to inform you that your manuscript has been deemed suitable for publication in PLOS ONE. Congratulations! Your manuscript is now being handed over to our production team.

Kind regards, 

on behalf of

Dr. Mayank Sukhija 

Academic Editor

PLOS ONE